# Apelin, a Circulating Biomarker in Cancer Evaluation: A Systematic Review

**DOI:** 10.3390/cancers14194656

**Published:** 2022-09-25

**Authors:** Christina Grinstead, Saunjoo Yoon

**Affiliations:** Department of Biobehavioral Nursing Science, College of Nursing, University of Florida, 1225 Center Drive, Gainesville, FL 32610-0187, USA

**Keywords:** apelin, cancer, tissue, serum, plasma

## Abstract

**Simple Summary:**

Despite advances in science and technology for the care of patients with cancer, providing effective treatment remains challenging. The lack of sensitive markers for early diagnosis and monitoring of cancer progression leads to suboptimal outcomes and decreased survival. There is a need to identify objective and reliable biomarkers that can be used in diagnosing, monitoring, and treating cancer. Apelin as a protein plays a role in cancer development and may predict treatment response and prognosis; however, research is limited. This review aims to synthesize current knowledge on associations of apelin circulating in blood with cancer and highlight knowledge gaps to direct future research. Results showed significant variations in methods of measuring apelin in the blood among various cancer studies and indicated inconsistent associations between apelin and clinical characteristics such as BMI, tumor pathology, and survival. Future research warrants more studies to advance in standardized measurement methods, produce a body of evidence based on the cancer types, and define the optimal cutoff points of apelin for accurate and early detection of cancer for personalized treatment.

**Abstract:**

Apelin is a promising biomarker for the detection and prognosis of cancer. This review aims to synthesize current knowledge on associations of circulating apelin with cancer, illustrate knowledge gaps, and discuss future research. Following PRISMA guidelines, CINAHL, EMBASE, and PubMed were searched using terms “cancer AND apelin” between 2011 and 2021, full text, and English language. Inclusion criteria: measured circulating apelin in adults 18 years or older with cancer, and observational, cross-sectional, longitudinal, case–control, cohort, quasi-experimental, or randomized control trials. Excluded were studies with animal models, tissue samples only, secondary data analyses, systematic reviews, literature reviews, grey literature, and conference abstracts. 16 articles were included. There were significant variations in measurement methods between studies. Comparison of circulating apelin between cases and controls and associations of circulating apelin with clinicopathological characteristics were inconsistent. Variations in results suggest that the relationship between circulating apelin and cancer differs among cancer types. Differences in measurement methods between studies highlight the need for consistency in future research to draw meaningful conclusions. Future research should seek to standardize methods of detecting circulating apelin and examine its associations with specific cancer types to determine what role that circulating apelin may play in cancer development and progression.

## 1. Introduction

Despite advances in science and technology for the care of patients with cancer, cancer remains a significant issue, being the second leading cause of death in the United States, with almost two million estimated new cases diagnosed and over half a million deaths projected this year [1]. Providing effective treatment remains challenging due to the lack of sensitive markers for early diagnosis and monitoring of cancer progression, leading to suboptimal outcomes and decreased survival. Identifying objective and reliable biomarkers for effective cancer screening and treatment is imperative in advancing personalized and precision medicine. Efforts to discover novel proteins and other molecules that may play a role in cancer development and progression have shown promising results [2,3]. Of such proteins, apelin has received attention in the scientific community since 1998 [4]. Apelin, named from ***AP***J ***E***ndogenous ***Li***ga***n***d, is an endogenous ligand for the G-protein coupled apelin receptor (APJ) that was first isolated from bovine stomach extract [4]. Research in the past decade has shown promise for apelin as a biomarker in cancer to predict treatment response and prognosis and detect cancer. This review aims to synthesize evidence on the association of circulating apelin with cancer, illustrate gaps in knowledge, and discuss potential methodological improvement in the more sensitive assessment of apelin levels and innovative approaches for future cancer research.

Apelin, synthesized as an immature single peptide of a 77-amino acid preproprotein, is found in multiple forms, with apelin-12, apelin-13, apelin-17, and apelin-36 the most biologically active. Apelin, found most often in the heart, lungs, and mammary glands, plays a role in many physiological processes such as apoptosis, inflammation, and tumor proliferation [5,6]. Apelin has implications in regulating body fluid homeostasis, glucose metabolism, energy metabolism, and neuroendocrine function, including the Hypothalamic-Pituitary-adrenal (HPA) axis [6]. It is commonly known to have an effect on the cardiovascular system with its inotropic properties and as a vasopressor and vasodilator [7]. It also promotes angiogenesis, cell migration, and cellular permeability [8].

Apelin may play a role in cancer development by activating the apelin receptor, APJ. Recent studies found APJ to be overexpressed in tumor tissues, particularly those metastasized [9,10,11]. While the exact molecular mechanisms behind the action of apelin through APJ are unknown, its role is likely in pathways such as STAT3, ERK, and AKT [9]. Such pathways associated with apelin and APJ activation promote tumorigenesis and metastasis through its cardiovascular effects [12,13]. Additionally, associations between circulating apelin and obesity suggest potential effects on chemotherapy response [14] and obesity-related cancers [5].

Despite a vast body of evidence examining apelin expression in the tissue, studies on circulating apelin in blood for cancer are scarce and have shown considerable variations in methodology, making it difficult to draw definitive conclusions. Additionally, there are significant gaps in knowledge on the role of circulating apelin in cancer. The source of circulating apelin is currently not well-known, and is suspected to differ between patient populations [15]. While previous research has shown associations between apelin and various pathologies, the exact mechanisms behind these associations remain unknown, particularly in cancer. This review summarizes available evidence on circulating apelin concentration and its association with clinicopathological characteristics, treatment, prognosis, and survival in cancer. Findings will provide information to explore the utility of circulating apelin as a biomarker for risk assessment, diagnosis, monitoring, and evaluation of cancer in clinical practice.

## 2. Methods

### 2.1. Literature Search Strategy

The systematic literature search followed Preferred Reporting Items for Systematic Reviews and Meta-Analysis (PRISMA) guidelines [16]. A preliminary search was conducted through 12 December 2021, using CINAHL, PubMed, and EMBASE. Search terms used included “cancer” AND “apelin”. Filters applied showed only studies published within 10 years of the date of the preliminary search, in English language, and in full text. Additional articles were identified by reviewing the reference lists of articles meeting the inclusion criteria.

### 2.2. Study Inclusion and Exclusion Criteria

Inclusion criteria were (1) measured serum or plasma apelin concentrations as a primary variable of interest in adults18 years of age or older with all cancer types, and (2) observational, cross-sectional, longitudinal, case–control, cohort, quasi-experimental, or randomized control trials. Exclusion criteria were studies with animal models, studies with tissue samples only, secondary data analyses, systematic reviews, literature reviews, grey literature, and conference abstracts.

### 2.3. Data Extraction and Quality Assessment

First, the abstracts of all articles were reviewed as the first screening process. Full texts of the selected abstracts were reviewed during the second screening. One independent reviewer organized articles meeting the inclusion criteria and manually performed data extraction. Extracted data included: First author, publication year, purpose, study design and methods, sample population, variables, and findings. Findings of interest included circulating apelin levels in patients with cancer and compared to controls when applicable, and associations between circulating apelin and demographics, clinicopathological characteristics, treatment modality, and survival. Additionally, noted were receiver-operating characteristic (ROC) analyses and comparisons between tissue and serum as a source for measuring apelin in cancer. All outcomes associated with findings of interest were collected, regardless of differences in study design such as measurement method, timepoints, and analysis.

Quality assessment was performed by one reviewer using the Observational Study Quality Evaluation (OSQE) assessment tool for case–control and cohort studies. The National Institutes of Health (NIH) Quality Assessment Tool for Before-After (Pre-Post) Studies with no control group was used for pre-post studies without the control groups. The OSQE for observational studies was created by combining relevant items from three commonly used and validated checklists [17]. The NIH quality assessment tools were developed by researchers at the NHLBI and Research Triangle Institute International (https://www.nhlbi.nih.gov/health-topics/study-quality-assessment-tools (accessed on 22 December 2021). For the OSQE tools, stars are awarded based on meeting specified criteria. The option to veto if one or more unmet criteria were deemed significant enough not to validate the study results. High-quality studies received ≥ 80%, moderate quality received 50% to <80%, and low quality received < 50% of possible stars. For the NIH quality assessment tools, the same cutoff values for the percentage of criteria were used to assign the high, moderate, or low quality.

## 3. Results

### 3.1. Search Results

The search results yielded 563 articles. Of those articles, 167 duplicate articles were removed before the screening. Due to irrelevance to this review, 271 articles were excluded after screening all abstracts. A full-text review was conducted for the remaining 124 articles. Of those 124, 108 articles were excluded as serum or plasma apelin was not a focus of the study, there were other than human subjects, the article was a review or from an excluded publication source, or the study population was under 18 years of old. A total of 16 articles were included in this review (Figure 1).

### 3.2. Study Design and Sampling

Of the 16 studies included, 11 were case–control, 4 were prospective case–cohort, and 1 was a pre-post test without a control group. Multiple cancer types were examined, including the gastrointestinal tract, lung, breast, gynecologic, prostate, multiple myeloma, non-Hodgkin’s lymphoma, head and neck cancer, and melanoma. For the case–control and cohort studies, 12 used healthy controls, 3 used non-healthy controls diagnosed with conditions affecting physiological processes related to the cancer being studied, and 1 lung cancer study used healthy non-smokers and non-patient smokers as controls. The pre- and post-test study included patients with cancer only (Table 1).

### 3.3. Risk of Bias

For the 16 studies included, the quality assessment indicated 1 high quality, 9 moderate quality, and 6 low quality evidence. Table 1 shows the study characteristics included in this review. The most common cause of the low-quality category was that the evidence did not provide sufficient information regarding the methodology. Other studies that scored low-quality raised concerns regarding the validity of their sampling methods when described and lacked reporting of relevant results. One study was vetoed because the evidence was low quality due to having a small sample size and low internal and external validity (Table 1).

### 3.4. Differences in Serum Apelin Levels among Cases and Controls

There was significant variation in the methodology for measuring apelin in the blood among the included studies. Twelve (12) studies measured serum apelin and four (4) studies with plasma apelin. Five studies that used serum did not provide information on the isoform measured, three studies measured apelin-12, and four studies measured apelin-36. Three studies that used plasma did not provide information on the isoform measured, and one study measured apelin-36. Assay quality varied widely, or no assay information was provided. There were significant differences in sensitivity, minimum detectible limit, and coefficients of variability for the studies that provided quality information on the assay used. Results were also reported in different units, including ng/mL, pg/mL, and ng/L.

Comparison of apelin levels between subsets of the study population showed wide variation among results. Most studies showed that apelin was higher in patients with cancer than in controls, while some showed lower apelin levels in cancer patients than controls. Additionally, not all studies reported statistically significant differences in apelin levels between patients with cancer and controls. The studies that compared cancer types or subtypes of a single cancer also showed varied results, with statistically significant differences in some but not all of the studies, indicating inconsistent findings among studies. Two studies did not provide numerical values for apelin concentration and the p-values for all results (Table 2).

### 3.5. Demographic and Clinical Characteristics and Apelin Concentration

Nine studies evaluated relationships between apelin concentration and demographics, including age and sex or gender. Sex or gender was not associated with apelin in five studies, but one study [21] showed that men had significantly higher serum apelin than women among obese patients with colon cancer. Age was not associated with apelin concentration in all five studies (Table 3).

Eight studies examined the relationships between apelin concentration and BMI and/or other body composition measurements (Table 3). Of those eight studies, four [18,19,31,32] reported no relationships between BMI or adipose tissue content and apelin concentration, while the other four studies [21,22,23,26] showed various levels of significant relationships, such as a positive correlation between BMI and apelin in three studies. One of these four studies [26] showed reduced apelin levels after 12 months had decreased waist-hip ratios and lower fat mass.

Ten studies looked at tumor characteristics and apelin. For cancer staging and classification systems used, TNM staging was the most common system and followed by Duke’s classification and FIGO staging. These studies showed inconsistent results, with some studies showing associations between stage, metastasis, histology, and tumor grade and size, and some finding no statistically significant associations. Two studies examined apelin and its association with treatment. One study showed no significant difference in apelin levels before and after radiotherapy, and one showed no significant change in apelin after treatment with chemotherapy (Table 3). Seven studies examined relationships between apelin concentration and a wide variety of blood-based parameters. The results are summarized in Table 4.

### 3.6. Associations between Apelin and Survival and Prognosis

Three studies [22,24] examined apelin levels associated with survival and prognosis among cancer patients. There were no associations between serum apelin and survival in non-Hodgkin’s lymphoma or gastric cancer. In comparison, two studies [22] reported statistically significant associations between apelin and survival and prognosis. Findings of the studies, however, were contradictory. For instance, cancer patients with disease progression or death within 24 months showed higher serum apelin at baseline [22], while multiple myeloma patients with higher serum apelin at the time of diagnosis had longer median survival [25].

### 3.7. Apelin Biomarker and Receiver Operating Characteristic (ROC) Curves Analysis

Studies examined associations between apelin and various characteristics, showing potential uses for apelin as a biomarker in cancer. All 16 studies looked at differences in apelin between cases and controls or different classifications of cancer types. Many studies also sought to determine associations between apelin and clinical characteristics, most often characteristics of tumor growth and metastasis. However, other factors such as BMI and additional cancer-relevant lab values were examined for associations with apelin.

Four studies analyzed ROC curves to find the optimal cutoff point of serum or plasma apelin in order to diagnose or monitor for disease progression of cancer patients [22,23,25,27]. The results of these four studies were highly inconsistent. The Area Under Curve (AUC) ranged from 0.676 to 0.97, and sensitivity and specificity were equally varied. Optimal cutoff points also were inconsistent (Table 5).

### 3.8. Serum vs. Tissue Apelin

Three studies compared serum apelin to tissue apelin [18,24,31]. These studies sampled both tumor and adjacent non-tumorous tissue from gastroesophageal, colorectal, and gastric carcinoma patients and found differences in tumor apelin expression compared to normal tissue. However, one study indicated that apelin tumor tissue expression was more closely associated with clinicopathological characteristics than serum apelin [24], while the two other studies showed no association between tumor apelin and clinicopathological characteristics [18,31]. All three studies found weak or no correlation between serum and tissue apelin [18,24,31].

## 4. Discussion

This review examined associations between serum apelin concentration, BMI, and body composition in cancer and clinicopathological characteristics related to tumor growth and metastasis. Results were mixed in associations between serum apelin and BMI in cancer. Previous research has shown apelin contributes to obesity-related disorders other than cancer, particularly those with insulin resistance [5]. For patients with cardiovascular disease, increased serum apelin was found in children and adults who were obese compared to the non-obese, especially those consuming a high-fat diet. Similarly, apelin levels decreased in persons with weight loss resulting from reduced fat and calorie intake [5]. In contrast, 4 of the 8 studies in this review that examined BMI and apelin reported no association. Few studies indicated positive associations between apelin and obesity-related cancers such as endometrial and breast cancer [22,23]. The 7 case–control and cohort studies did not measure apelin changes over time, so it is not clear if a causal relationship exists between serum apelin and the development of cancer. It is conceivable that patients who develop obesity-related cancers such as breast or colorectal may have increased serum apelin due to obesity, not from the presence of cancer [23].

When examining patients with early stage hormone receptor-positive breast cancer, serum apelin levels and the BMI were higher in patients with breast cancer compared to the age and BMI-matched healthy control group (*p* = 0.0001) [26]. Within-group comparison in breast cancer patients between the increased and the decreased apelin groups indicated that decreased apelin group had significantly increased waist and hip ratios (*p* = 0.008) and increased fat mass (*p* = 0.047) than the counterpart at the 12-month follow-up [26]. Further studies warrant unveiling the association of the decreased serum apelin level with cancer progression and fat mass. It will be essential to consider confounding factors affecting serum apelin levels in obesity-related cancers to delineate changes in apelin and the underlying mechanisms associated with cancer development and progression.

Most studies included in this review examined associations between serum apelin and tumor characteristics such as stage and metastasis. Results, however, were mixed from no association to association of serum apelin with increased in later stages of cancer and those with lymph node and distant metastasis. Previous research reported that tissue apelin concentration correlated with tumor growth and metastasis [5,33] suggesting the vascular effects of apelin. Apelin has been shown to affect blood pressure and control the vascular tone in non-cancer cases [8]. The apelin-APJ signaling system also induces cell migration. In cancer, it is hypothesized that these properties induce tumor growth and metastasis by promoting angiogenesis, cell proliferation, and cell permeability. This hypothesis is supported by a significant body of research showing increased apelin concentration and overexpression of the APJ receptor for which apelin is a ligand in tumor tissue [5,10]. It is unclear whether the relationship between serum apelin concentration and cancer cell proliferation and migration is similar to apelin found in tumor tissue.

Results indicate the potential use of apelin as a biomarker for various cancers. Every study looked at differences in apelin between either cases and controls or different subtypes of cancer, showing that apelin may be of use in cancer diagnosis. Additionally, apelin studies examining associations with clinical characteristics may be used to determine whether these associations may be strong enough to predict outcomes such as the likelihood of metastasis or development of signs and symptoms commonly found in those with shorter survival or worse prognosis. Our review supports a potential association, but it is too early to conclude the definitive associations because of insufficient evidence and conflicting study findings. We also hypothesize that the association of apelin may differ with various cancer types. Further research is needed to evaluate the utility of serum apelin as a prognostic marker for cancer progression.

Serum apelin may be beneficial in the diagnosis of cancer. Several studies included in this review compared serum apelin concentration between patients with cancer and healthy controls or among different cancer types. Although results were inconsistent, most studies showed serum apelin increased in cancer compared to controls. In general, studies reported conflicting results among different cancer types. It was noticeable that three studies concluded inconsistent results among similar cancer types by showing increased serum apelin in two cancer studies [18,31] and one showing no difference between cases and controls in those with gastric and/or esophageal cancers [24]. Four studies of ROC analysis showed vastly different optimal cutoff points of serum apelin concentration to differentiate patients with cancer from controls, ranging from 0.160 ng/mL to 6.85 ng/mL. A synthesis of these results may not be appropriate due to the heterogeneity of cancer types and until a sufficient body of evidence for individual cancer classification is compiled to use as a biomarker in cancer diagnosis and to determine and produce results to apply in clinical practice.

Tissue apelin levels have been studied extensively, with some conflicting results. However, tissue apelin seems to be correlated with clinicopathological characteristics [18,24]. However, while tumor tissue samples are often not readily available, serum samples are less invasive and potentially more helpful for measuring apelin concentration in many cases such as advanced cancers. Additionally, serum apelin may be helpful in screening, whereas measures of tissue apelin are often only applicable after identifying the tumor. The three studies included in this review showed weak or no correlation between serum apelin and tissue apelin [18,24,31], but associations between serum apelin with cancer were shown in two of the studies. Serum apelin levels did not differ between patients with gastric cancer and the control group with chronic gastritis [24]. Conversely, there were associations between serum apelin and several clinicopathological characteristics in colorectal cancer [31] and higher serum apelin levels in gastroesophageal cancer than in healthy controls; however, apelin showed no significant associations with tumor characteristics [18]. Serum apelin and tissue apelin concentration show inconsistent associations with the development and progression of cancer, and yet serum apelin still has potential for use in cancer diagnosis and prognostic monitoring as a biomarker separate from tissue apelin. More research will help understand the relationship between serum and tissue apelin; however, their uses in cancer may be unrelated.

This review examined current evidence as to the relevance of blood-based measures of apelin to cancer. Our review indicated inconsistency of isoform measures from measuring different isoforms of the apelin protein to providing no information about types of isoforms. This has implications as to whether any sound conclusion of the association between serum apelin and cancer development and progression may be made based on the evidence available. A recent study has shown that pry-13 apelin and apelin-17 are the major isoforms found in circulation, although concentrations of circulating isoforms differ between study populations [15]. It suggests that research should focus on identifying the isoform of apelin most relevant to the study population. Variations in lab techniques, processes, and challenges in sample analysis due to the short half-life of apelin may cause difficulties in maintaining consistency among studies. Our review demonstrated significant variations in apelin levels between cases (experimental group) and controls, as well as ROC analysis. Measurement methods should be consistent and refined for a definitive conclusion and meaningful clinical application of the findings.

Our review has multiple strengths. We identified the gaps in current research and provided recommendations for future directions through a comprehensive review. We highlighted issues that may potentially prevent meaningful synthesis, providing information for future researchers to consider when designing studies to have the most significant impact on the current body of existing knowledge while promoting the efficient use of often limited time and resources.

There are limitations to this review. Most studies were cross-sectional in nature and so did not examine changes in apelin over time. Results should be interpreted cautiously because they often had a moderate or high risk of bias due to study design and lack of reporting. Another limitation is the heterogeneity of cancer types and methodologies and the limited number of studies, which makes it challenging to draw a conclusion in detail by cancer types. The last limitation is potentially missing information from published articles in 2022 if any since the literature search was completed at the end of 2021.

## 5. Conclusions

There has been increased attention on apelin for cancer research since 1998. Despite an increased number of studies and advances in methodologies, current research on serum apelin and its associations with cancer show conflicting results. More research is needed to investigate the role of serum apelin in cancer development, progression, and predictive value. Serum apelin concentration is likely to vary based on the differing metabolic and physiologic processes among patients with different cancer types. It further highlights the need to identify isoforms relevant to specific cancer types as it is likely that their concentrations will differ between patient populations. Future research should seek to produce a sufficient body of evidence on each cancer type individually to conclude whether changes in serum apelin concentration over time can be associated with specific disease processes. In elucidating these processes, it will be essential to consider confounding factors affecting serum apelin concentration for a more definitive conclusion. For use in clinical settings, serum apelin shows promise in furthering attempts to improve screening practices and provide individualized care for patients diagnosed with cancer. It is critical to define the optimal cutoff points of serum apelin concentration indicating the presence of disease and its related processes to identify patients with cancer and implement treatment earlier, improving health-related outcomes and overall survival.

## Figures and Tables

**Figure 1 cancers-14-04656-f001:**
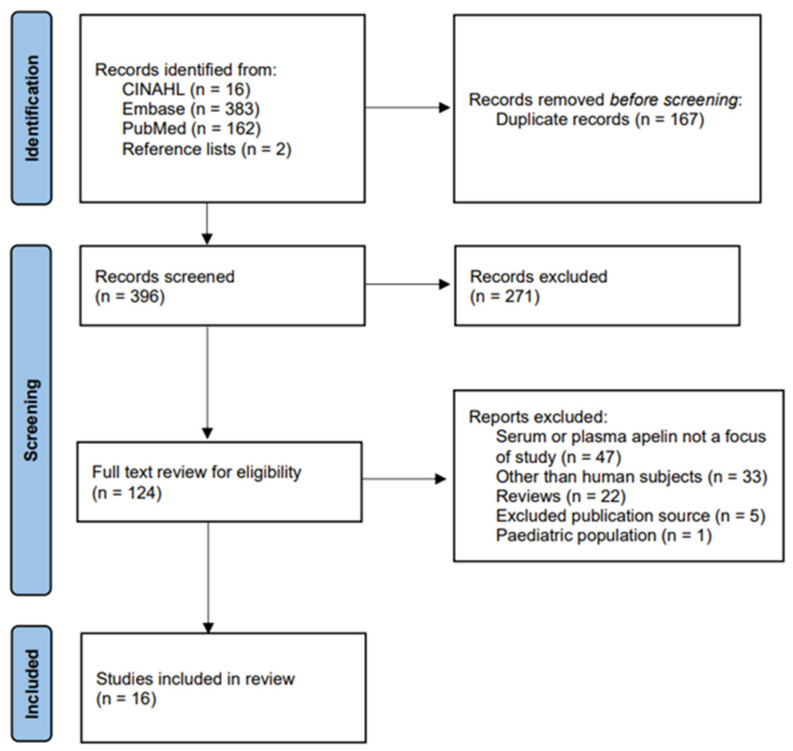
PRISMA diagram [16].

**Table 1 cancers-14-04656-t001:** Summary Description of Study Characteristics.

Author/Year Study/Design Location	Purpose	Sample	Quality Assessment
Diakowska, et al., 2014 [18] Case–control Poland	To investigate the relationship between cachexia syndrome and serum resistin, adiponectin, and apelin in patients with gastroesophageal cancer	145 Total 85 Gastroesophageal Cancer 39 Squamous cell carcinoma of the esophagus 22 Adenocarcinoma of the gastroesophageal junction 24 Gastric adenocarcinoma 60 healthy controls	9/15 stars 0 vetoes Moderate
Tulubas, et al., 2014 [19] Case–control Turkey	To investigate the association of serum adipokines with colon cancer and bowel adenomas	90 Total 30 Colon Cancer 30 Bowel adenomas 30 healthy controls	10/15 stars 0 vetoes Moderate
Yang, et al., 2014 [20] Case–control China	To investigate the role of apelin in the cell proliferation and autophagy of lung adenocarcinoma	20 Total 10 Lung cancer 10 Healthy controls	6/14 stars 2 vetoes Low
Al-Harithy, et al., 2015 [21] Case–control Saudi Arabia	To evaluate serum concentrations of apelin-12 and its relationship to the metabolic profile of patients with colon cancer	119 total 59 colon cancer 60 healthy controls	11/15 stars 0 vetoes Moderate
Lacquaniti, et al., 2015 [22] Prospective case–cohort Italy	To examine the correlations between apelin expression and clinical outcomes in oncologic patients, such as cancer disease progression and patient survival	130 Total 95 All cancers 32 Lung cancer 23 Gastrointestinal cancers 9 Breast and Gynecologic cancers 14 prostate cancer 35 healthy controls	14/22 stars 0 vetoes Moderate
Altinkaya, et al., 2015 [23] Case–control Turkey	To investigate serum apelin levels in women with endometrial cancer and correlate apelin levels with tumor markers, metabolic profiles, and clinicopathologic features	90 Total 46 Endometrial cancer 44 controls with uterine dysfunction	12/14 stars 0 Vetoes High
Feng, et al., 2016 [24] Prospective case–cohort China	To study the association between apelin expression and the clinical features and postoperative prognosis in patients with gastric cancer	351 Total 270 Gastric cancer 81 Controls with chronic gastritis	10/19 stars 0 vetoes Low
Maden, et al., 2016 [25] Prospective case–cohort Turkey	To examine the association between apelin levels and clinical findings in multiple myeloma (MM) and Non-Hodgkin’s Lymphoma (NHL)	79 Total 29 Multiple Myeloma 31 Non-Hodgkin’s Lymphoma 19 Healthy controls	12/20 stars 0 vetoes Moderate
Salman, et al., 2016 [26] Case–control Turkey	To investigate changes in apelin levels in postmenopausal Breast Cancer patients receiving aromatase inhibitors	60 Total 40 Breast cancer 20 Healthy controls	10/22 stars 0 vetoes Low
Ni, et al., 2017 [27] Case–control China	To evaluate whether circulating apelin could act as a biomarker in lung cancer diagnosis	186 Total 129 Lung cancer 57 Healthy controls	10/15 stars 0 vetoes Moderate
Aktan, et al., 2019 [28] Prospective case–cohort Turkey	To investigate whether there was a variety in serum apelin levels between patients with head and neck cancer (HNC) and a healthy control group and to compare the serum apelin levels before and after radiotherapy	52 Total 22 Head and Neck Cancer 30 Healthy controls	10/22 stars 0 vetoes Low
Diakowska, et al., 2019 [29] Case–control Poland	To evaluate the mRNA expression and protein levels of apelin, the apelin receptor, resistin, and adiponectin in esophageal squamous cell carcinoma	80 total 53 cases of Esophageal Squamous Cell Carcinoma 20 Cachexia 33 No cachexia 27 healthy controls	10/14 stars 0 vetoes Moderate
Gholamnejad, et al., 2019 [30] Case–control Iran	To evaluate the amount of serum apelin in smokers without cancer and the correlation of apelin with the types of lung cancer found in smokers	124 Total 4 Non-smoker lung patients 59 smoker lung cancer patients 12 adenocarcinoma 22 squamous cell carcinoma 8 small cell carcinoma 17 other types of carcinoma 61 Controls - 30 Smokers - 31 Non-smokers	7/15 stars 0 vetoes Low
Podgόrska, et al., 2019 [31] Case–control Poland	To perform a quantitative analysis of apelin and its receptor expression in colorectal cancer patients	83 total: 56 patients with colorectal cancer 27 healthy controls	8/15 stars 0 vetoes Moderate
Berta, et al., 2021 [10] Case–control Sweden	To investigate the role of apelin in the growth of melanoma lung metastasis	61 total 40 patients with melanoma 21 healthy controls	9/15 stars 0 vetoes Moderate
Grupińska, et al., 2021 [32] Pre-post test without control Poland	To investigate levels of biomarkers associated with oxidative stress and adipokines in breast cancer patients with different clinical features and analyze the effect of chemotherapy on their concentrations.	60 total 60 patients with breast cancer	6/8 criteria 3 unknown Low

Note. Quality Assessment checklist used was the Observational Study Quality Evaluation (OSQE) for case–control and case–cohort studies [17]. The NIH quality assessment tool for before-after pre-post test studies without control was used for pre-post test studies without controls (https://www.nhlbi.nih.gov/health-topics/study-quality-assessment-tools (accessed on 22 December 2021)).

**Table 2 cancers-14-04656-t002:** Summary Description of findings on Apelin levels in case controls and cohorts.

Author (Year)	Apelin Measure	Quality of Assay	Group (n) Comparisons	Apelin Levels (ng/mL or pg/mL or ng/L)	(*p* < 0.5)
Diakowska, et al., 2014 [18]	Serum Unknown isoform	Sensitivity:0.09 ng/mL Intraassay CV: 5–10% Inter-assay CV: <15%	GEC/cachexia (44): HC (60) GEC/non-cachectic (41): HC (60) ESCC (39): GADCA (46)	855 ± 195: 635 ± 365 820 ± 211: 635 ± 365 886 ± 127: 836 ± 118 pg/mL	*p* = 0.014 ns ns
Tulubas, et al., 2014 [19]	Serum Unknown isoform	Minimum detectible level: 0.6 ng/mL	Colon Adenoma (30): HC (30): CC (30)	2.88 ± 0.48: 2.98 ± 0.66: 1.63 ± 0.37 ng/mL	*p* < 0.000
Yang, et al., 2014 [20]	Plasma Unknown isoform	No information provided	Lung cancer (20): HC (10)	370:200 ng/mL	*p* < 0.05
Al-Harithy, et al., 2015 [21]	Serum Apelin-12	Apelin-12 enzyme immunoassay kit Analytical range: 0.0–100 ng/mL	Obese male CC: Non-obese CC: Obese control Obese controls: Obese female CC	0.85: 0.35/0.29: 0.72/0.57 0.72/0.57: 0.26 ng/mL	
Lacquaniti, et al., 2015 [22]	Serum Unknown Isoform	No information provided	Baseline-Various cancers (95): HC (30) Cancer progression after 24 mths: Cancer progression (41): No progression (54)	532.5: 231 717.4: 392.1 pg/mL	*p* < 0.0001
Altinkaya, et al., 2015 [23]	Serum Apelin-36	Intraassay CV: <10%, Interassay CV: <12% Detection range: 37.04–3000 pg/mL	Endometrial cancer (46): HC (44)	215.1: 177.3 pg/mL	*p* < 0.002
Feng, et al., 2016 [24]	Serum Apelin-12	Sensitivity: 0.05 ng/mL Intraassay CV: <5% Interassay CV: <10%	Gastric cancer: Chronic gastritis	2.84 ± 1.13: 2.52 ± 0.78 ng/mL	ns
Maden, et al., 2016 [25]	Plasma Unknown isoform	No information provided	MM (29): NHL (31) MM (29): HC (19) NHL (31): HC (19)	1.99: 0.56 1.99: 0.42 0.56: 0.42 ng/mL	*p* < 0.001 *p* < 0.001 ns
Salman, et al., 2016 [26]	Serum Apelin-36	Interassay CV: <12% Intraassay CV: <10%	Breast cancer (40): HC (20)	N/A	*p* < 0.0001
Ni, et al., 2017 [27]	Plasma Unknown isoform	No information provided	HC (57): Lung Cancer (129)	750: 550 ng/mL	*p* < 0.0001
Aktan, et al., 2019 [28]	Serum Apelin-36	Interassay CV: <12% Intraassay CV: <10% Minimum detection limit: 0.07 ng/mL Analytical range: 0–100 ng/mL	HNC (22): HC (30) Pre-RT (22): Post-RT (22)	1.66: 0.53 1.66: 1.51 ng/mL	*p* < 0.001 ns
Diakowska, et al., 2019 [29]	Serum Unknown isoform	sensitivity: 2.63 pg/mL intra-assay CV: <10%, inter-assay CV: <12%	ESCC (53): HC (27) Ca w/ cachexia (20): Ca w/o cachexia (33)	746.7: 570 907.6: 671.7 pg/mL	*p* = 0.036 *p* = 0.006
Gholamnejad, et al., 2019 [30]	Serum Apelin-12	sensitivity: 4.47 ng/L, assay range: 10–4000 ng/L, intraassay CV: <8%, inter-assay CV: <10%	Lung SCC: Lung ADCA: SCLC: Other lung carcinoma Smokers w/no cancer: Smokers w/ lung cancer	2205.54: 1088.00: 797.25: 1000.37 2142.20: 1461.92 ng/L	*p* < 0.05 *p* < 0.05
Podgόrska, et al., 2019 [31]	Serum Apelin-36	Sensitivity: 2.63 pg/mL	CRC (56): HC (27)	N/A	
Berta, et al., 2021 [10]	Plasma Apelin-36	No information provided	Melanoma (40): HC (21)	1.46: 0.77 ng/mL	*p* = 0.0011
Grupińska, et al., 2021 [32]	Serum Unknown isoform	No information provided	Breast ca (60) HER2+ (20):HER2-–(40)	1.58:1.16 ng/mL	*p* = 0.002

Note. MM = Multiple Myeloma. NHL = Non-Hodgkin’s Lymphoma. GEC = Gastroesophageal cancer. HC = Healthy Controls. ESCC = Esophageal squamous cell carcinoma. GADCA = Gastric adenocarcinoma. CC = Colon cancer. HNC = Head and neck cancer.

**Table 3 cancers-14-04656-t003:** Summary of Relationships Between Demographic and Clinical Characteristics and Apelin.

Author/Year	Data of Interest Collected	Key Findings
**Diakowska, et al., 2014 [18]**	Serum apelin Demographics BMI Tumor histology and TNM staging	Serum apelin is not correlated with sex, age, BMI Serum apelin is not associated with tumor histology, TNM stage, tumor stage (T), lymph node metastasis, or distant metastasis
**Tulubas, et al., 2014 [19]**	Serum apelin BMI	No correlation between serum apelin and BMI
**Al-Harithy, et al., 2015 [21]**	Serum apelin Duke’s classification, metastasis, tumor size Demographics BMI/anthropomorphic measurements	Obese males show an inverse relationship between serum apelin and stages Obese men with colon cancer with higher serum apelin than obese women with colon cancer In non-obese males, serum apelin correlated with BMI
**Lacquaniti, et al., 2015 [22]**	Serum apelin Demographics BMI TNM staging	Apelin is positively correlated with BMI, but not correlated with gender. Serum apelin differs between staging groups, increasing with increasing stage, and between those with and without metastases.
**Altinkaya, et al., 2015 [23]**	Serum apelin Demographics BMI Tumor characteristics	Apelin is not associated with age Serum apelin is higher in obese compared to non-obese women with endometrial cancer; in healthy controls, apelin levels of obese compared to non-obese were similar Apelin positively correlated with BMI Serum apelin inversely correlated with FIGO stage but not tumor grade or size
**Feng, et al., 2016 [24]**	Serum apelin Demographics Tumor size, differentiation, TNM stage, lymph node metastasis, distant metastasis	Serum apelin is not correlated with gender or age Serum apelin is higher in cases with lymph node metastasis, but is not correlated with tumor size, tumor differentiation, and distant metastasis.
**Maden, et al., 2016 [25]**	Plasma apelin Tumor staging	NHL showed a negative correlation between apelin and stage, apelin decreased in advanced stages
**Salman, et al., 2016 [26]**	Serum apelin BMI/Body composition	Cases with reduced apelin levels after 12 months of follow-up had decreased waist-hip ratio and fat mass compared to those with higher levels
**Ni, et al., 2017 [27]**	Plasma apelin Demographics Histological tumor classification, stage, metastasis	Serum apelin is lower in cases with metastasis compared to controls No association of apelin with histological classification, tumor stage, age, and gender
**Aktan, et al., 2019 [28]**	Serum apelin Treatment	No significant difference in apelin levels before and after radiotherapy
**Diakowska, et al., 2019 [29]**	Serum apelin Demographics TNM staging, tumor size, lymph node metastasis, distant metastasis	No association between serum apelin and age and gender Apelin is positively associated with TNM stage, but not associated with lymph node metastasis, distant metastasis or tumor size
**Podgórska, et al., 2019 [31]**	Serum apelin BMI Tumor Characteristics	No association between serum apelin and BMI Serum apelin is higher in those with more advanced TNM stages and those with lymph node metastasis and distant metastasis
**Berta, et al., 2021 [10]**	Plasma Apelin Demographics	No correlation between apelin level and age
**Grupińska, et al., 2021 [32]**	Serum apelin Demographics BMI Tumor characteristics Treatment	No correlation of apelin with age, BMI, or adipose tissue contentApelin is not associated with tumor size, histopathological grade, and lymph node metastasis No significant change to apelin after treatment with chemotherapy

Note. BMI = Body Mass Index. TNM = primary tumor, lymph node, metastasis. FIGO = Fédération Internationale de Gynécologie et d’Obstétrique staging system. MM = Multiple Myeloma. NHL = Non-Hodgkin’s Lymphoma.

**Table 4 cancers-14-04656-t004:** Relationships Between Blood-Based Measures and Serum Apelin.

		First Author (Year)						
	Diakowska, et al., 2014 [18]	Tulubas, et al., 2014 [19]	Al-Harithy, et al., 2015 [21]	Lacquaniti, et al., 2015 [22]		Altinkaya, et al., 2015 [23]	Salman, et al., 2016 [26]	Maden, et. al., 2016 [25]
**Variables/Cancer**	Gastroesophageal	Colon	Colon	Multiple		Endometrial	Breast	MM/NHL
**Hemoglobin**	-			+ *	0 **			+ ^NHL^
**Lymphocytes**	0							
**Thrombocytes**								0
**Total Protein**	0	0						
**Albumin**	0			+ *	0 **			
**hsCRP**	+							
**Creatinine**								0
**eGFR**				+				
**Uric Acid**				- *	0 **			
**LDH**								- ^MM^
**Total Cholesterol**		0	U	+ *	0 **	0	+	
**Triglycerides**		0	0			0	0	
**HDL**		0	0			0		
**LDL**		0	U			0		
**Fasting insulin**						+		
**Fasting glucose**		0	0			0		
**Phosphate**				- *	0 **			
**Potassium**				- *	0 **			
**Sodium**				-				
**Calcium**								0
**IgM**								- ^MM^
**IgA**								0 ^MM^
**IgG**								0 ^MM^

Note. + = positive association or correlation with serum apelin; - = negative association or correlation with serum apelin; 0 = no correlation or association with serum apelin; U = correlation found but no information on direction provided; shaded squares indicate data was not collected on the parameter or its association with serum apelin was not analyzed. Where results of analysis differed within the same study: * = results from univariate analysis. ** = results from multivariate analysis. +^NHL^ = results for Non-Hodgkin’s Lymphoma patients. - ^MM^ = results for Multiple Myeloma patients. Studies not listed in this table did not measure or analyze any parameters included in this table. MM = Multiple Myeloma. NHL = Non-Hodgkin’s Lymphoma. LDH = lactic dehydrogenase. hsCRP = high-sensitivity C-reactive protein. eGFR = estimated glomerular infiltration rate. IgM = Immunoglobulin M. HDL = High density lipoprotein. LDL = Low density lipoprotein. K = Potassium. Na = Sodium. Ca = Calcium. IgA = Immunoglobulin A. IgG = Immunoglobulin G.

**Table 5 cancers-14-04656-t005:** Summary of results of ROC analysis.

Author/Year	Purpose of Analysis	Optimal Cutoff Concentration	AUC	Sensitivity	Specificity
Lacquaniti, et al., 2015 [22]	To find the best cutoff values capable of identifying the progression of oncologic disease	Serum Apelin unknown isoform 610 pg/mL	0.97 (95% CI: 0.91–0.99)	92.7	94.4
Altinkaya, et al., 2015 [23]	To find the optimal cutoff points of apelin measurements for diagnosis of endometrial cancer	Serum Apelin-36 160 pg/mL	0.676 (95%CI: 0.567–0.786)	87.0	43.2
Maden, et al., 2016 [25]	To find the optimal cutoff point for diagnosis of MM	Plasma Apelin unknown isoform 0.827 ng/mL	0.842 (95%CI: 0.739–0.945)	76	86
Ni, et al., 2017 [27]	To find the optimal cutoff point for diagnosis of lung cancer	Plasma Apelin unknown isoform 6.85 ng/mL	77.7 (CI not provided)	84.2	71.4

## Data Availability

All data generated or analyzed during this study are included in this published article.

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
