# Peer review of "Apelin, a Circulating Biomarker in Cancer Evaluation: A Systematic Review"

_cancers, 2022, doi:10.3390/cancers14194656_

Round 1

Reviewer 1 Report

The authors submitted a systematic review in which they summarizes available evidence on circulating apelin concentration and its association with clinicopathological characteristics, treatment, prognosis, and survival in cancer. They screened English language written articles in the CINAHL, PubMed, and EMBASE and finally included 16 article in the review. The authors did not establish significant difference in the levels of circulating apelin between cases and controls and associations of circulating ape-
lin with clinicopathological characteristics in cancer patiens. The aim of the study is clear and consice. The manuscript has a logical structure and contains well-balaced subsections that cover all aspects of the study. The study was provided in accordance to
g PRISMA guideline. The findings seem to be out of manipulation. The tables are legible and clear. The discussion is regarded to be comprehensive and redable. The conclusive section is attractive for readers. Although the findings are intriguing, I would like to put forward several comments to discuss.

1. The authors should clearly report what molecular mechanisms link apelin synthesis and release with cancer progression,

2. Whether hystology (for instance, adenocarcinoma) is a prominant cause leading to apelin isoform changes ?

3. The authors accumulate the findings received from various study in which the cancer status of the patients was variable. If this was a reason for the finding of the study?

Reviewer 2 Report

The systematic review “Apelin, a circulating biomarker in cancer evaluation: A systematic review” is a very well-written, thorough summary and analysis of relevant literature. The manuscript describes the literature search, review and summary of findings for each included article as well as a comprehensive description of excluded articles. Among the findings were that the different studies used different methods for detecting apelin, measured different isoforms of apelin and studied different patient populations. 

The conclusions were that future studies should standardize the methods and the measured isoform of apelin, establish the relationship between circulating apelin and type of cancer as well as how apelin is involved in cancer development, progression, metastasis and prognosis. Further studies are needed to establish the relevant isoform in a particular study population and determine how circulating apelin concentrations correlate with cancer type and localization. 

The selection of literature was carefully made and comprehensively described and the conclusions relevant. In order to improve the manuscript further, under the heading “Serum vs. Tissue Apelin” it would be nice if the authors could describe in more detail what type of tissue samples are referred to here. Biopsies? Resections to remove tumor? Healthy tissue adjacent to tumor?
